# Variational Inference for Laser Disturbance Detection in Powder Bed Fusion

## Abstract

In this study we use variational inference to learn a dynamics model from a high-speed video stream of a laser melting process. We compare two deep generative sequence models and evaluate them on video prediction and anomaly detection tasks. We find that the latent representation provides sufficient robustness to detect anomalies to high levels of performance (AUROC=0.9999). The method is generally applicable to high dimensional time-series modelling and distils the temporal data-stream to a single metric.

## 1 Introduction

Digital manufacturing methods such as 3D printing can benefit greatly from generative models. In particular powder bed fusion contains multi-scale physics which is a challenge to simulate effectively (Wei et al., 2021). However, terabytes of in-situ data can be collected to gain insights into the process (Clijsters et al., 2014; Hooper, 2018). This is needed since a random disturbances can cause small voids to form within the material. Hence, process monitoring is important for adoption in industries such as aerospace, medical and nuclear which require high quality components (Clijsters et al., 2014). Detecting such defects in-situ would further advance the technology by allowing control and repair systems to be activated.

However, the cost of collecting anomalous data can be significant. There may be requirements for destructive testing which can be expensive. Diversity of anomalies results in intractability of such modelling in many instances since anomalous events emerge in an unpredictable manner. The nature of an anomaly also means it is, in general, less likely to be observed resulting in unbalanced data during inference which can also be a challenge. Hence, an approach is required to both robustly detect, and quantify anomalies.

In an anomaly detection task, a new data point is compared to prior knowledge which can be represented by a generative model. Probabilistic approaches to deep learning such as the Variational Autoencoder (VAE) enable generative models of a dataset to be learned (Kingma & Welling, 2013; Rezende et al., 2014). Similarly, temporal equivalents allow dynamics to be learned from structured data such as speech generation and action recognition (Bayer & Osendorfer, 2014; Chung et al., 2015). In this paper we explore the use of approximate variational inference to learn the dynamics of a laser imaged from high-speed video sensor data. We quantify performance of the learned dynamics models on a video prediction task. We then evaluate each model's ability to detect anomalous process signatures.

## 2 Problem Description

In the powder bed fusion process, a laser repeatedly melts thin layers of metal powder one after the other until a 3D component is formed (Fig. 1). This enables complex, light weight, high strength components. However, defects can form within the part which is a challenge to inspect. In this problem, we wish to define the distribution of normal processing conditions by learning a generative model of the data. This has physical meaning since the laser operates in a controlled environment and therefore has predefined behaviour governed by physical laws. If the energy density drops below a threshold, anomalous processing occurs. This can cause material defects to form and hence, results in the need for costly post-inspection procedures which contribute to large portions of the component cost.

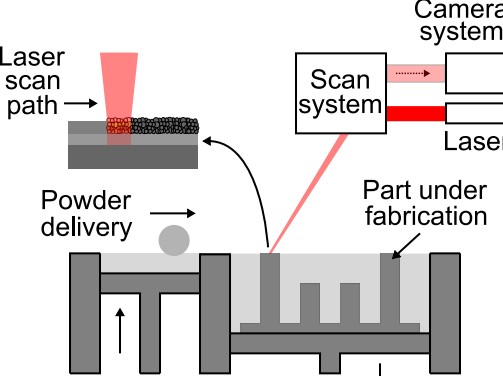

Figure 1: Schematic of the laser powder bed fusion process with camera coaxial to the scan system optics.

The alternative approach would be to collect data and classify the anomalies directly without a generative model. The problem arises that this data is prohibitively expensive and time-consuming to collect. Similarly, should an instrument or a material change, the data collection would need to be repeated. Another approach would be to apply multi-scale simulation to generate the distribution and then convert the sensor data to the simulated domain. However, simulation costs are time-consuming taking many orders of magnitude longer when compared to data collection to achieve equivalent fidelity (Wei et al., 2021).

To detect a disturbance or unknown event, an anomaly detection methodology can be adopted. An anomalous or outlier observation may be defined loosely as a data point which differs considerably from other data points within a set of samples (Grubbs, 1969). In the context of this application, the process has a generating distribution, $\mathbf{x} \sim p(\mathbf{x}_t | \mathbf{z}_{\leq t}, \mathbf{x}_{<t})$, where $\mathbf{x}_t$ is the observation at time, $t$, sampled from latent variables, $\mathbf{z}$. Since the data points are images, it is hypothesised that comparing the new data points to the latent distribution, $p(\mathbf{z}_t | \mathbf{x}_{\leq t})$ is more effective in this application since small deviations need to be detected. Hence three things need to be determined: (1) an evaluation of the generative model, (2) a distance metric to compare new samples and, (3) an appropriate threshold.

Two datasets were captured for the purposes of training and testing the models. The laser parameters were adjusted to create regions of optimal and anomalous laser melting. This resulted in training dataset of 175,000 frames and test set of 175,000 frames keeping experiments independent. These were split into sequences of 20 frames resulting in 17,500 sequences in total. Images were cropped to $64 \times 64$ pixels around the expected laser position.

## 3 METHODOLOGY

To learn the generative model we compare two sequence models, Stochastic Recurrent Networks (STORN) (Bayer & Osendorfer, 2014) and the Variational Recurrent Neural Network (VRNN) (Chung et al., 2015).

The STORN model optimises the following variational lower bound:

$$\mathcal{L} = \mathbb{E}_{q(\mathbf{z}_{\leq T} \,|\, \mathbf{x}_{\leq T})} \Big[ \sum_{t=1}^{T} \big( - D_{\mathrm{KL}}(q(\mathbf{z}_t \,|\, \mathbf{x}_{\leq t}) \| p(\mathbf{z}_t)) + \log p(\mathbf{x}_t \,|\, \mathbf{z}_{\leq t}, \mathbf{x}_{<t}) \big) \Big]. \tag{1}$$

The VRNN was introduced to build additional dependencies between the latent variables. This was achieved by introducing a non-gaussian prior. The objective function is a step-wise lower bound:

$$\mathcal{L} = \mathbb{E}_{q(\mathbf{z}_{\leq T} \mid \mathbf{x}_{\leq T})} \Big[ \sum_{t=1}^{T} \big( - D_{\mathrm{KL}}(q(\mathbf{z}_t \mid \mathbf{x}_{\leq t}, \mathbf{z}_{<t}) \| p(\mathbf{z}_t \mid \mathbf{x}_{<t}, \mathbf{z}_{<t})) + \log p(\mathbf{x}_t \mid \mathbf{z}_{\leq t}, \mathbf{x}_{<t})) \Big]. \quad (2)$$

In both cases, the inference and generative models are learned jointly by maximising the lower bounds in Eq. (1) and (2).

Since our task is video based we use a convolutional neural network (CNN) to embed images into latent space and transposed convolutions as the decoder. The implemented architecture follows Burgess et al. (2018).

It is well known that generative models do not perform as expected on data drawn from novel unseen distributions (Louizos & Welling, 2017; Nalisnick et al., 2018). To overcome this, the model is optimised on both in-distribution and outlier data[1]. However, only the in-distribution data is penalised with the Kullback-Leibler (KL) divergence from Eq. (1) and (2).

## 4 EXPERIMENTS

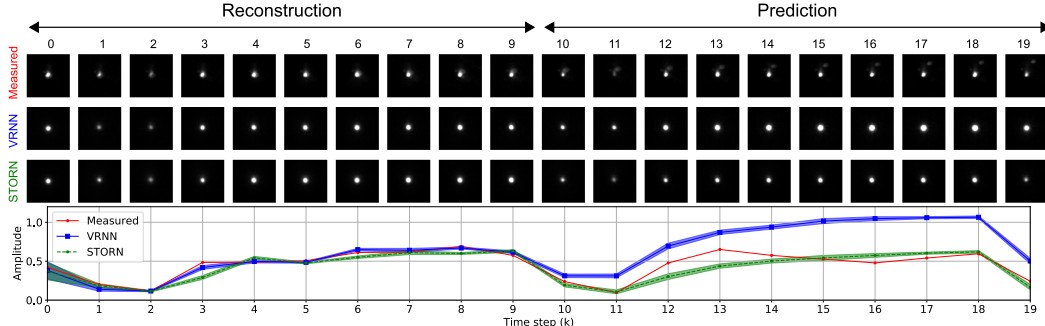

Figure 2: Assessment of the video prediction task on an anomalous sample. The ground truth observation is shown in the first row. Row two and three show the VRNN and STORN respectively. The plot shows corresponding peak image brightness at each time step. The first 10 frames are reconstructed, the last 10 are predicted.

### 4.1 VIDEO PREDICTION

In this section predictive dynamics are quantified to understand which model has a better learned representation. A similar procedure is followed to Srivastava et al. (2015), where future prediction is quantified by measuring error on the predicted images to evaluate the learned representations. We therefore evaluate the models on multi-step prediction. As input, the model sees the first ten frames and predicts the next ten frames. A qualitative comparison between models can be seen in Fig. 2. We report the mean squared error (MSE) prediction of the 10 frames in Table 1.

The models are capable of learning complex temporal dependencies across time steps. This includes both temporal and spatial structure. The STORN model is better at prediction and generalises better to anomalous data illustrated by the lower MSE scores. It can be seen in Fig. 2 that the VRNN tends to follow optimal dynamics when making predictions, while STORN generalises better. There is also a trade-off between making predictions and reconstruction. As the model learns better representations, these become harder to predict. Therefore, managing this trade-off needs to be understood in terms of the requirements of the problem.

---

[1]Recent previous work by the authors, citation hidden to preserve anonymity during peer review.

Table 1: Performance evaluation of 10 frame prediction compared to reconstruction (RC).

| Model | MSE RC optimal | MSE pred. optimal | MSE RC anom. | MSE pred. anom. |
|-------|---------------|-------------------|--------------|-----------------|
| VRNN  | 0.333         | 0.505             | 0.309        | 5.888           |
| STORN | 0.331         | 0.450             | 0.350        | 0.587           |

Table 2: AUROC for anomaly detection features extracted from each model. The score is based on thresholding a single metric.

| Model | 1-step KL | 10-step KL | Mahalanobis | MSE pred. | MSE RC |
|-------|-----------|------------|-------------|-----------|--------|
| VRNN  | 0.9999    | 0.9999     | 0.8643      | 0.9918    | 0.0110 |
| STORN | 0.9994    | 0.9998     | 0.9999      | 0.0383    | 0.0063 |

## 4.2 ANOMALY DETECTION

In the case of anomaly detection, we wish to detect whether an image sequence contains an anomaly or not. Several different scores were used to identify these. Firstly, one step predictive KL divergence was used from Eq. (1) and (2). Secondly, we compare this with the predictive as well as the reconstructed MSE. Finally, we explore the use of a statistical distance measure, namely the Mahalanobis distance, which is a well known method of detecting out-of-distribution data in neural networks (Ren et al., 2021).

The results from two models can be seen in Table 2. Both models perform similarly in terms of the area under receiver operating characteristic (AUROC). The KL metrics are very robust and suggest that detection of anomalies in the latent space is effective. The Mahalanobis distance is less effective for the VRNN. However, the reconstruction feature does not perform well for either model.

The MSE predictor is less effective for the STORN model. On further analysis, we find that, due to STORN's better predictive ability, it is more sensitive to subtle dynamics changes in the latent space. We show one such example in Fig. 2, where a sequence has a minor dynamic disturbance. Hence, the VRNN is less sensitive in the case of dynamic disturbances and tends to predict the optimal image brightness. Further, investigation will be required to understand how to balance this sensitivity so that excessive false positives do not occur. This is especially a challenge in the anomaly detection framework, where anomalies are rare.

## 5 CONCLUSION AND FUTURE WORK

This study showed the applicability of variational inference to modelling the complex dynamics in powder bed fusion, where the models are learned from raw sensor data. We showed that squared error can be used to evaluate learned dynamics on future prediction. We also showed that the reconstruction term is not useful for anomaly detection purposes in this application. The detection of anomalies in latent space was highly effective and suggest a robust approach to anomaly detection. The approach is relevant for structured high-dimensional data in cases where unknown spatio-temporal dynamics exist and actionable information is required.

The choice of prior in generative models is clearly important, as can be seen by the differences between the models. A limitation of our work is that the physics are not accounted for and need to be learned from scratch. Tailoring the prior to the problem would likely provide added benefit, especially to long term predictions, and may be a useful avenue of future research.

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
