# OpenReview forum: "Variational Inference for Laser Disturbance Detection in Powder Bed Fusion"
_ICLR.cc/2022/Workshop/DGM4HSD — Submitted to ICLR 2022 DGM4HSD workshop_

### Official Review · Reviewer_yBR8 · 2022-03-18
**Review of Variational Inference for Laser Disturbance Detection in Powder Bed Fusion**

**Rating:** 4
**Confidence:** 4

**Review:**

The paper proposes applying generative model setup to detect anomalies during laser melting, The problem area is extremely interesting and can be a good fit for the workshop. However, I have concerns with the experimental setup.

* The paper has a setup where they generate real and anomalous laser melting data. For this particular setup, the paper needs to provide more information both qualitatively and quantitatively on what differentiates anomalous vs real laser melting data. Further, if this information is available a priori as in this particular setup, a classifier would be a better alternative.
* The authors suggest that only the in-distribution data is penalised with the Kullback-Leibler (KL) divergence from Eq. (1) and (2). Why is this the case?
* The high numbers in Table 2 are because the model is trained on both real and anomalous data, while specifically encouraging the posterior distribution of the labeled anomalous data to be far away from the prior. for the anomalous data. This is done by disabling the KL term. The authors should either
   a) train a baseline classifier to differentiate anomalous vs real data or
   b) train models on “real data” only and evaluate anomalies in test time.

Minor:
* These were split into sequences of 20 frames resulting in 17,500 sequences in total. It should be 10 frames instead of 20?
 * This was achieved by introducing a non-gaussian prior. I think the prior is still gaussian but conditioned on temporal information.

---

### Official Review · Reviewer_rgDv · 2022-03-20
**Very preliminary work applying generative modeling applied to detect anomalies in automated manufacturing**

**Rating:** 5
**Confidence:** 3

**Review:**

The authors report an attempt to apply generative modeling to detect anomalies in an automated manufacturing process.

Strengths:
* seems like an innovative application to an otherwise neglected domain in ML
* the manuscript provides a fairly accessible background on the application domain (though see weaknesses)
* it seems like a significant amount of effort was put in to collect experimental data (though see weaknesses)

Weaknesses and suggestions for improvement:
* though the manuscript is well motivated, it leaves gaps for the reader to fill up -- e.g. (1) it isn't clear how the experimental data was collected or what an anomaly 'looks' like; which ties into (2) the size and resolution of the images in Figure 2 don't reveal what a reader should be looking for; (3) In section 4.1, they say "the VRNN tends to follow optimal dynamics" but it isn't clear what "optimal dynamics" here mean.
* Since the manuscript is about an application (rather than building new methods), I feel like the authors’ focus on the actual value of evaluation metrics (AUROC=0.999) is misguided, and should be replaced with clearly defining why these metrics are relevant to the application at hand. More effort should be made on connecting or differentiating these metrics from similar metrics in related domains using longitudinal image data.

In summary, I think the authors’ work fits the theme of the workshop and might be of interest to other attendees, but the manuscript definitely requires significant work even for a workshop.

---

### Decision · Program_Chairs · 2022-03-26

**Decision:**

Reject

**Comment:**

Both reviewers agree that the topic of the submission is very interesting. However, consistent concerns were raised about the exposition on the acquired data, and what constitutes an anomaly. The AC encourages the authors to take into account the reviewer's suggestions and resubmit to a future venue.